# Lipid droplets and ferritin heavy chain: a devilish liaison in human cancer cell radioresistance

Luca Tirinato[1,2]*[†], Maria Grazia Marafioti[1,2][†], Francesca Pagliari[1]*[†], Jeannette Jansen[1,3], Ilenia Aversa[1,2], Rachel Hanley[1], Clelia Nisticò[1,2], Daniel Garcia-Calderón[1,3], Geraldine Genard[1], Joana Filipa Guerreiro[4], Francesco Saverio Costanzo[2], Joao Seco[1,3]*

[1]Biomedical Physics in Radiation Oncology, German Cancer Research Center (DKFZ), Im Neuenheimer Feld, Heidelberg, Germany; [2]Experimental and Clinical Medicine Department, University Magna Graecia of Catanzaro, Catanzaro, Italy; [3]Department of Physics and Astronomy, Heidelberg University, Im Neuenheimer Feld, Heidelberg, Germany; [4]Centro de Ciências e Tecnologias Nucleares, Instituto Superior Técnico, Universidade de Lisboa, Lisboa, Portugal

*For correspondence:
tirinato@unicz.it (LT);
f.pagliari@dkfz.de (FP);
j.seco@dkfz.de (JS)

[†]These authors contributed equally to this work

Competing interest: The authors declare that no competing interests exist.

**Abstract** Although much progress has been made in cancer treatment, the molecular mechanisms underlying cancer radioresistance (RR) as well as the biological signatures of radioresistant cancer cells still need to be clarified. In this regard, we discovered that breast, bladder, lung, neuro-glioma, and prostate 6 Gy X-ray resistant cancer cells were characterized by an increase of lipid droplet (LD) number and that the cells containing highest LDs showed the highest clonogenic potential after irradiation. Moreover, we observed that LD content was tightly connected with the iron metabolism and in particular with the presence of the ferritin heavy chain (FTH1). In fact, breast and lung cancer cells silenced for the FTH1 gene showed a reduction in the LD numbers and, by consequence, became radiosensitive. FTH1 overexpression as well as iron-chelating treatment by Deferoxamine were able to restore the LD amount and RR. Overall, these results provide evidence of a novel mechanism behind RR in which LDs and FTH1 are tightly connected to each other, a synergistic effect that might be worth deeply investigating in order to make cancer cells more radiosensitive and improve the efficacy of radiation treatments.

## Introduction

Since its first application in cancer treatment, radiotherapy has greatly improved from both a technical and a bio-clinical point of view, significantly increasing the treatment options and patient survival. Ionizing radiations (X-rays) work by damaging cell biomolecules, mostly DNA, which eventually induce cell death. The molecular mechanisms activated by cancer cells in response to ionizing radiation are extensively investigated and many advances have been so far made, but considerably many questions are still unanswered and much remains poorly understood. Cancer cell radioresistance (RR) makes different tumor types difficult to treat. In this regard, the presence within the tumor mass of a small cell subpopulation called cancer stem cells (CSCs) or cancer-initiating cells (CICs) seems to represent one of the driving forces contributing to tumor resistance and recurrence after radiotherapy treatments (*Baumann et al., 2008*).

Recently, lipid metabolic reprogramming in cancer cells has become a central aspect of cancer aggressiveness (*Cruz et al., 2020*; *Tirinato et al., 2017*). In particular, an increase of small lipid organelles inside cancer cells, namely lipid droplets (LDs), has been shown to correlate with a CSC

phenotype in colon (*Tirinato, 2015*) ovary (*Li, 2017*), breast (*Havas, 2017*), and glioblastoma (*Kant et al., 2020*).

Cell survival upon radiation treatment is also modulated by several tumor parameters such as hypoxia, oxidative stress, inflammation, acidic stress, and low glucose, all of which have been reported to mediate their effects through iron metabolism (*Schonberg, 2015*).

To date, altered expression and activity of many iron-related proteins in cancer cells have been reported and associated to cancer progression and metastasis (*Torti and Torti, 2013*; *Wang et al., 2010*). In fact, an uncontrolled balance of iron results in the free radical production through the Fenton reaction ($Fe^{2+} + H_2O_2 \rightarrow Fe^{3+} + \bullet OH + OH^-$), and free radicals are considered strong contributors to tumor proliferation and aggressiveness (*Bystrom et al., 2014*). Among all molecules involved in iron metabolism, ferritin is responsible for the cytoplasmic iron storage and the maintenance of the redox homeostasis. Ferritin is a protein complex composed of two chains, light (FTL) and heavy (FTH), and its clinical importance has been demonstrated in many cancers through multiple roles: the contribution to tumorigenesis, the restoration of tumor-dependent vessel growth, and the association with tissue invasion (*Schonberg, 2015*; *Aversa et al., 2017*). Moreover, high levels of ferritin are often found in patients with various advanced cancers, which could potentially be treated with radiotherapy (*Lee et al., 2019*), although iron homeostasis is still poorly investigated in the context of radiation oncology.

A recently published paper highlighted a very intriguing relationship between iron balance and LDs. The Authors showed that iron depletion caused endoplasmic reticulum (ER) expansion and, as a consequence, LD accumulation into the cytoplasm of breast cancer cells (*De Bortoli et al., 2018*). These findings prompted us to investigate the LD role and the potential connections between FTH1, and indirectly iron balance, and LDs in various X-ray-treated cancer cells with the aim at identifying possible shared features, which can be targeted and manipulated to sensitize cells to the treatments.

This study demonstrates that radioresistant cancer cells of different origins (neuroglioma, lung, breast, bladder, and prostate) were characterized by a higher expression of LDs. The subpopulation containing the highest amount of LDs (LD^High) showed a higher clonogenic potential compared to the LD^Low counterpart. Interestingly, the number of cytoplasmic LDs was directly correlated with the amount of FTH1.

Altogether, these data provide evidence of a new pivotal role for LDs in cancer RR linking their expression with iron metabolism and specifically to FTH1 expression.

## Results
### X-ray radiation treatment enhances LDs
To verify whether LD content was affected by ionizing radiation treatment, H4 (neuroglioma), H460 (lung), MCF7 (breast), PC3 (prostate), and T24 (bladder) cancer cells were treated with 6 Gy X-ray and left in culture for 72 hours (hrs) in order to select only surviving and resistant cells. Treated and untreated cancer cells were stained with LD540 and imaged at the confocal microscope for the detection of LDs.

As shown by z-projection confocal microscopy images, surviving cancer cells were characterized by a significant increase of LDs for all the aforementioned cell lines (*Figure 1A*). Although the LD increase was a common feature observed in all cell lines, the relative LD ratio between treated and untreated cells showed little differences, with H460 exhibiting the highest amount (*Figure 1B*). LD modulation after radiation was further investigated at the gene level. Perilipin (PLIN) genes code for the proteins associated with LD surface and they are involved in their biogenesis as well as in several other roles (*Kimmel and Sztalryd, 2016*). Differences in tissue expression have been reported for all the PLIN genes (PLIN 1–5). Accordingly, we observed that, after 6 Gy radiation treatment, PLIN1 was upregulated in H460, MCF7, PC3, and T24; PLIN2 was downregulated in H460; PLIN3 showed mRNA increased expression in H4 and MCF7; PLIN4 expression was incremented in MCF7 and T24; PLIN5 resulted downregulated in MCF7 and upregulated in PC3 and T24.

It is well known that photon radiation acts, at the molecular level, producing reactive oxygen species (ROS) (*Schonberg, 2015*). In this regard, we found that cytoplasmatic ROS, measured by means of fluorogenic CM-H2DCFDA probe, were significantly upregulated in H4, H460, MCF7, and PC3, while no differences were detected in T24, after radiation (*Figure 1—figure supplement 1A*).

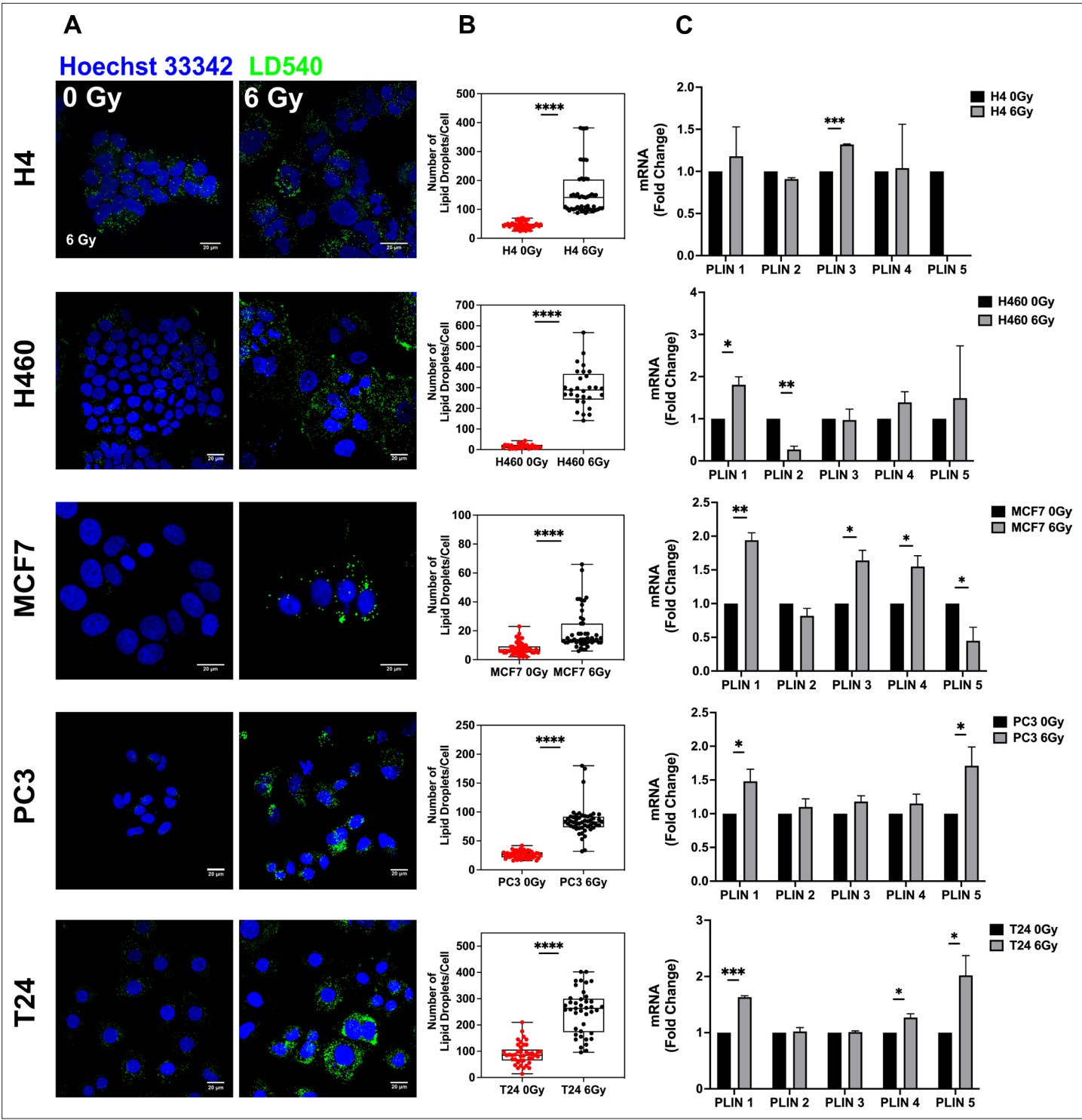

**Figure 1.** Lipid droplet detection in neuroglioma (H4), lung (H460), breast (MCF7), prostate (PC3), and bladder (T24) 6 Gy X-ray-resistant cancer cells. Cancer cells have been irradiated with 6 Gy X-ray and left in culture for 72 hrs. Afterwards, surviving and untreated cells have been stained with LD540 and imaged at the fluorescence confocal microscope. Z-projection of the z-stack acquisitions for untreated and 6 Gy treated cells are reported in column (**A**) (Scale bar, 20 μM). (**B**) For each cell line, 50 cells have been randomly imaged, and their LD number counted by using FiJi software. (**C**) qPCR analysis of the PLIN genes in the indicated cell lines. PLIN5 in the H4 6 Gy treated cells is not reported in the graph because it was not expressed. Error bars represent the means ± SD from three independent experiments. *≤0.05; **≤0.01; ***≤0.001, and ****≤0.0001.

The online version of this article includes the following figure supplement(s) for figure 1:

**Figure supplement 1.** ROS evaluation in 6 Gy radioresistant cancer cells.

*Figure 1 continued*

**Figure supplement 2.** ROS and Lipid Droplet Double Staining in Different Cancer Cell Lines.

**Figure supplement 3.** CSC Marker Evaluation in 6 Gy Radioresistant Cells.

Moreover, H4, H460, and PC3 showed upregulated levels of SOD1, SOD2, and catalase, respectively. SOD2 mRNA was also upregulated in T24, despite the fact that general ROS levels were not altered 72 hrs after radiation treatment, while SOD expression was downregulated in radiation-treated MCF7. (*Figure 1—figure supplement 1B*).

In order to deal with this ROS increase, cancer cells need to tune their ROS scavenging systems (*Trachootham et al., 2009*), and among all scavenging systems, LDs have been observed to contribute to the modulation of excessive oxidative stress (*Welte, 2015*). Furthermore, by co-staining LDs and ROS in the heterogeneous not-irradiated cancer populations, we found that populations with higher LDs also exhibited higher levels of ROS (*Figure 1—figure supplement 2*). Therefore, LD content, influencing cell survival, was directly correlated with ROS production in all cell lines.

Previous works reported that ionizing radiation could selectively enrich the cancer cell population of cells with stem-like properties (*Ghisolfi et al., 2012*; *Krause et al., 2017*; *Woodward et al., 2007*). Thus, we have analyzed the expression of some of the most common markers used to identify CSCs. In particular, we found that CD44 was upregulated in 6 Gy treated H4, H460, MCF7 and T24; CD133 mRNA increased in H4 and H460-irradiated cells; CD166 expression was upregulated in MCF7 and T24; ALDH1 was incremented in MCF7. On the contrary, PC3 RR cells did not display significant increase in the expression of such CSC markers (*Figure 1—figure supplement 3*).

## LD$^{high}$ subpopulation retains the highest clonogenic potential

LD modulation following X-ray treatment raised the question if LD accumulation was a consequence of radiation treatment or if such a feature was already present in some cells within the heterogeneous cancer populations, thus suggesting that LD content could participate in conferring a higher radiation resistance.

To better define the role played by LDs in RR cells and to address the question, H4, H460, MCF7, PC3, and T24 were stained with LD540, sorted in the 10 % highest and lowest LD-containing cells (LD$^{High}$ and LD$^{Low}$ cells) (*Figure 2*) and, soon after, irradiated with 2, 4, and 6 Gy X-ray. The surviving fractions (SFs), calculated for all cell lines at the different doses, showed that LD$^{High}$ cells retained the highest clonogenic potential and therefore they were the most radioresistant (*Figure 2*). These results suggest that the LD amount present in the population is linked to a stronger cell capability to survive ionizing radiations, independently of the tissue of origin.

## Ferritin heavy chain (FTH1) affects LD accumulation and cell radio-response

One of the main cellular ROS sources is the Fenton reaction, in which the $Fe^{2+}$ reacts with hydrogen peroxide ($H_2O_2$) to produce $Fe^{3+}$ and highly reactive radicals, such as the hydroxyl radical ($\cdot OH$). Since the Ferritin is the main intracellular iron storage protein, we investigated the FTH1 role in radiation resistance.

We found that FTH1 protein was upregulated in all resistant cell lines after 72 hrs from 6 Gy exposure, as reported in *Figure 3A,B*. Moreover, H460 and MCF7, sorted on the basis of their LD content, were characterized by an increase in the mRNA level of FTH1 in the LD$^{high}$ subpopulation compared to the LD$^{low}$ cells (*Figure 3C*).

To better clarify this link, FTH1 silencing in H460 and MCF7 (H460$^{shFTH1}$ and MCF7$^{shFTH1}$), the efficiency of which is shown in *Figure 3D*, was performed. FTH1 silencing resulted in influencing cell ability to deal with free cytoplasmic iron, as demonstrated by the downregulation of Transferrin Receptor 1 (TfR1) mRNA and the upregulation of Ferroportin (FPN) mRNA, all involved in proper iron homeostasis (*Figure 3—figure supplement 1A,B*).

Interestingly, in FTH1 silenced H460 and MCF7, the amount of FTH1 directly correlated with the number of LDs (*Figure 3E,F*). In fact, H460 $^{shFTH1}$ and MCF$^{shFTH1}$ cells were characterized by a significant reduction of LDs. This, in turn, led to an evident increase in the radiosensitivity, as demonstrated by the clonogenic assay results (*Figure 3G*).

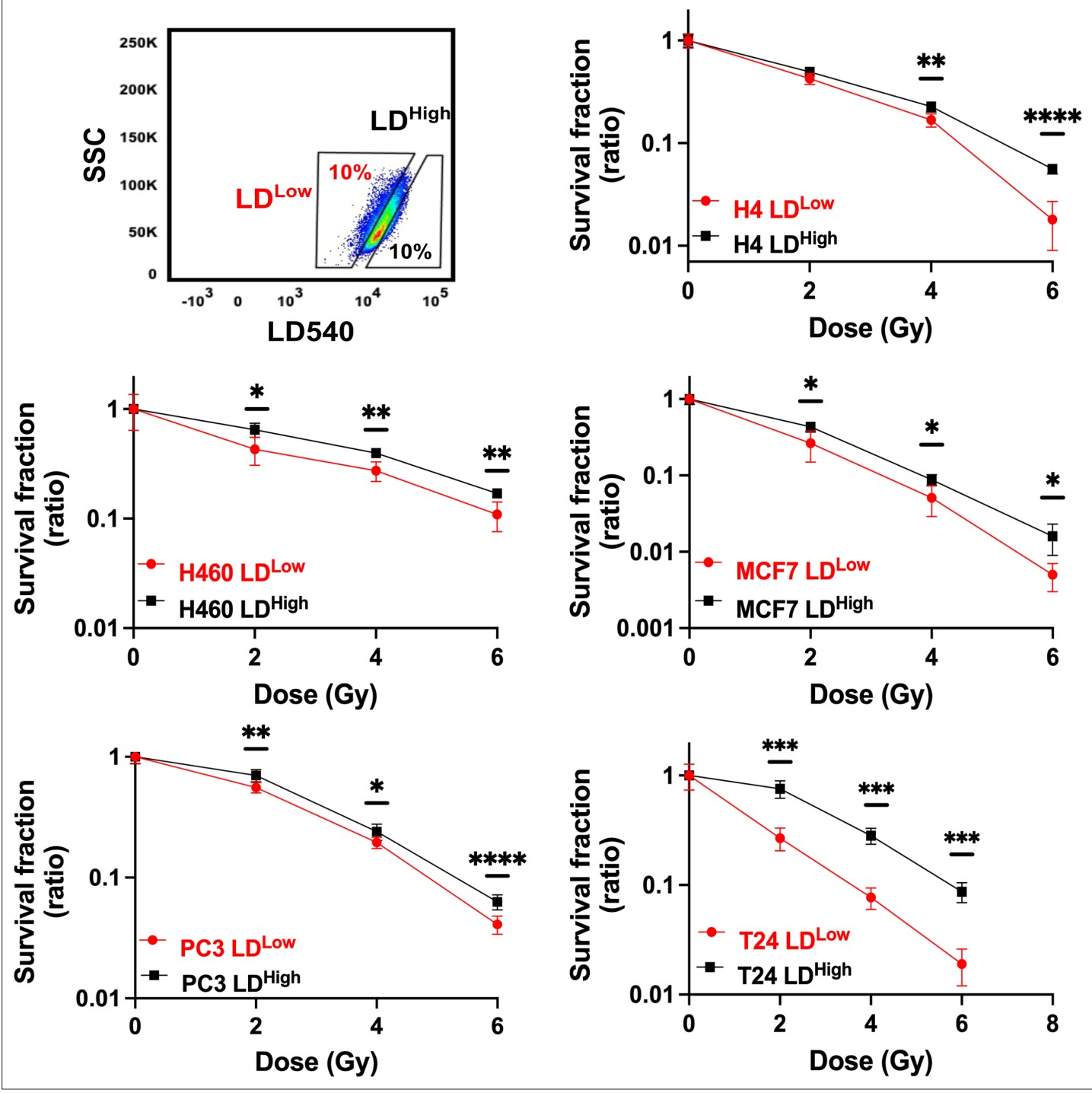

**Figure 2.** Cell survival curves for H4, H460, MCF7, PC3, and T24 cancer cell lines. All cancer cell lines were stained with LD540 and then sorted in the 10 % highest and lowest LD-containing cells (box up-left). For each cell line, the two LD sub-populations were irradiated at 2, 4, and 6 Gy X-ray and their survival fraction calculated. Survival fractions are reported in log-linear scale. Error bar represents the means ± SD from three independent experiments. *≤0.05; **≤0.01; ***≤0.001 , and ****≤0.0001.

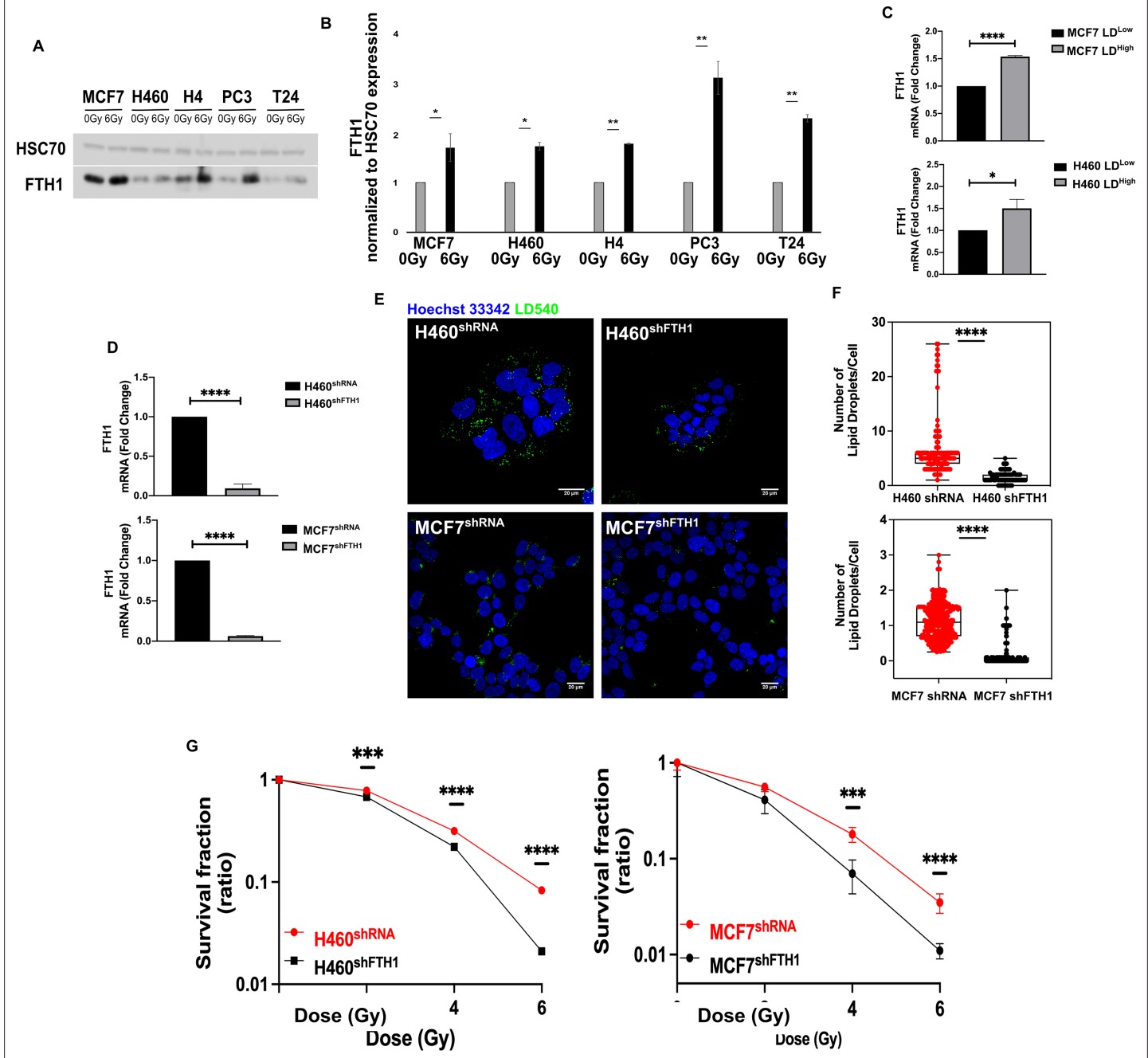

**Figure 3.** FTH1 silencing downregulates lipid droplets affecting cancer radioresistance. (**A, B**) Wester blotting analysis and quantification of FTH1 expression in MCF7−, H460−, H4−, PC3−, T24- 0 Gy vs 6 Gy X-ray-treated cells. HSC70 was used as a loading control. (**C**) H460 and MCF7 were sorted in the 10 % Highest (H460 LD$^{High}$ and MCF7 LD$^{High}$) and Lowest (H460 LD$^{Low}$ and MCF7 LD$^{Low}$) LD-containing cells and then FTH1 mRNA expression measured by qRT-PCR in all four sub-populations. Primer sequences are listed in Key resource table. (**D**) H460 and MCF7 were silenced for FTH1 by lentiviral-driven shRNA strategy. PCR results showed that in H460 shFTH1 and MCF shFTH1 there was a clear FTH1 mRNA reduction compared with their relative controls. (**E, F**) LD content was measured and quantified in H460 shFTH1 and MCF7 shFTH1 by confocal microscopy. LD540 staining revealed that the FTH1 gene silencing caused a LD decrease in both cell systems. (Scale bars 20 µM). (**G**) Cellular irradiation response in H460 and MCF7 silenced for FTH1 was investigated by radiobiological clonogenic assay and compared with H460 shRNA and MCF7 shRNA, respectively. Survival fraction (in log-linear scale) is reported in (**G**). Error bar represents the means ± SD from three independent experiments. *≤0.05; **≤0.01; ***≤0.001, and ****≤0.0001.

The online version of this article includes the following figure supplement(s) for figure 3:

**Figure supplement 1.** TfR1 downregulation and FPN upregulation in FTH1-silenced H460 and MCF7 cells.

Summarizing, we show that LD content was dependent on the FTH1 expression and thus linked to the free cytoplasmic iron. When the levels of the main protein responsible for iron storage decreased, LDs were also reduced and this significantly impaired cancer RR.

## Iron imbalance as well as FTH1 reconstitution re-establish LD expression and radiation resistance

As well known, the FTH1 role is crucial for the iron storage within the cell and the maintenance of the redox homeostasis. When its expression is downregulated, the balance between the iron uptake and release is compromised. By consequence, the free cellular iron amount becomes critical for the correct cellular functions (*Wang et al., 2010*; *Rouault, 2006*).

Here we found that this iron imbalance assumed also a central role in the LD accumulation. Given the role played by the FTH1 deficiency on LD content and radiosensitivity, we overexpressed the FTH1 by FTH1 cDNA transfection to further verify such connection. *Figure 4A,B* shows that FTH1 protein was successfully raised up in both MCF7 and shFTH1 + pcDNA$_3$FTH1 and H460 and shFTH1 + pcDNA$_3$FTH1.

Moreover, such an overexpression fully restored the LD pool (*Figure 4C–D*) in both cell lines, which also reacquired a higher RR (*Figure 4E*).

To further elucidate the connection between iron and LDs, we used an iron chelator agent, Deferoxamine (DFO), to cope with the iron imbalance due to the FTH1 silencing. DFO is a high-affinity $Fe^{3+}$ chelator and an FDA approved drug used to treat patients with iron overload. H460$^{shFTH1}$ and MCF7$^{shFTH1}$ were treated with DFO for 24 hrs, and their LD content was analyzed. LD540 staining on both treated H460$^{shFTH1}$ (H460 shFTH1+ DFO) and MCF7 shFTH1 (MCF7 shFTH1+ DFO) univocally showed that the iron chelation was able to induce LD accumulation (*Figure 4F–G*), and this, in turn, conferred higher survival ability to cells after radiation treatment, as shown by clonogenic assays (*Figure 4H*).

## Discussion

Along with surgery and chemotherapy, radiotherapy represents an important treatment option also in the palliative regimens. Great advance in the understanding of the molecular mechanisms underlying the cancer RR have been done. Nevertheless, this has not translated into a proportional improvement of the therapeutic outcomes because multiple biological factors and their complex interactions capable of negatively affecting the cellular response to ionizing radiation remain to be characterized. RR exhibited by many cancer cells, especially CSCs, severely limits the effectiveness of the treatments. For this reason, the identification of distinctive features for targeting RR cells is critical, and it is currently the focus of intense research. Classical CSC markers used to identify the most putative RR cells are still being debated due the high intra- and inter-tumor heterogeneity (*Shackleton et al., 2009*) and the cancer cell ability to change during cancer progression and treatments. In recent years, accumulating studies suggested that LDs might be correlated with a CSC phenotype and an elevated tumorigenic potential (*Tirinato, 2015*). Increased LD amount has been found in various cancer cells with stem-like properties, including colorectal cancer cells (*Tirinato, 2015*), glioblastoma cells (*Kant et al., 2020*), and breast cancer cells (*Havas, 2017*).

In the present study, lung (H460), neuroglioma (H4), breast (MCF7), prostate (PC3), and bladder (T24) cancer cells were irradiated with 6 Gy X-rays and left in culture for 3 days in order to select only the RR cells. Surviving cells from all cell lines exhibited an elevated LD content, to which corresponded a cell type-dependent upregulation of the PLIN genes, whose proteins play a role in the formation and structure of LDs. These RR cells also showed differential and cell-specific upregulation of some CSC markers in almost all cell lines. Although we did not screen all the putative stemness markers, our preliminary data indicates that radiation exposure might enrich the heterogeneous population with cells having a stemness-like phenotype, which is in agreement with previous works (*Ghisolfi et al., 2012*; *Woodward et al., 2007*).

Meanwhile, RR cells with high LD content showed higher ROS levels associated with an increased antioxidant ability, as demonstrated by the genetic upregulation of antioxidant scavenging enzymes, such as SOD and Catalase. However, this behavior was not common to all cell lines, and, in fact, in T24 ROS levels remained unchanged. This suggest that cells from different origin were able to deal with

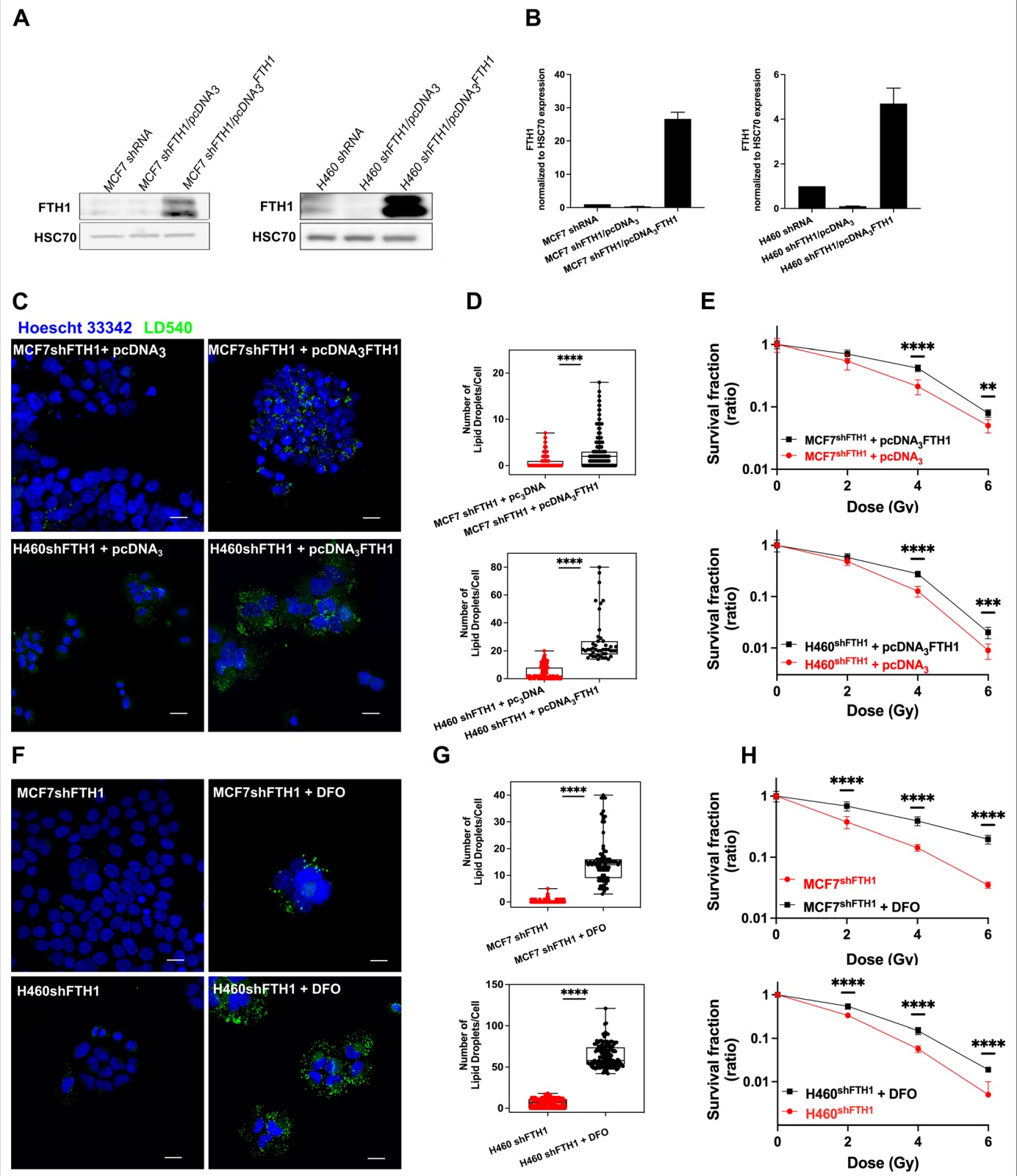

**Figure 4.** FTH1 reconstitution as well as DFO treatment restore the LD content re-establishing cancer radioresistance. (**A-B**) Western Blotting analysis and quantification of FTH1 expression in MCF7 shFTH1/pcDNA3FTH1 and H460 shFTH1/pcDNA3FTH1. HSC70 was used as loading control. (**C**) Z-stack representative confocal fluorescence images of LD detection and quantification (**D**) in MCF7 shFTH1/pcDNA3FTH1 and H460 shFTH1/pcDNA3FTH1 cells and their H460 shFTH1/pcDNA3 and MCF7 shFTH1/pcDNA3 controls. (Scale bars 20 mM). (**E**) Survival fractions (in log-linear scale) after FTH1

*Figure 4 continued on next page*

*Figure 4 continued*

reconstitution in MCF7- and H460- shFTH1 cells. (**F**) Z-stack representative confocal fluorescence images of LD detection and quantification (**G**) in MCF7 shFTH1 and H460 shFTH1 treated with DFO (Scale bar 20 mM). (**H**) Survival curves (in log-linear scale) of FTH1-silenced MCF7 and H460 cells after DFO treatment. F. Error bar represents the means ± SD from three independent experiments. *£ 0.05; **£ 0.01; ***£ 0.001 and ****£ 0.0001.

The online version of this article includes the following source data for figure 4:

**Source data 1.** Uncropped blots for *Figure 4A*.

the high dose radiation in different ways, but they all shared the ability to accumulate cytoplasmic LDs. Of note, the presence of cells with high levels of ROS in the not-irradiated cells also displaying high levels of LDs suggest that LDs could serve an antioxidant system being able to buffer the excess of ROS. Indeed, high ROS levels are commonly found in many cancer cells and LDs could contribute to create a tolerable oxidative microenvironment and to better counteract the excess of ROS produced by irradiation.

In support of that, we demonstrated that a higher LD content was a feature already present in the heterogeneous populations, as pre-sorted (LD$^{high}$ and LD$^{low}$) cells displayed differential survival capacity after radiation, with the LD$^{high}$ subpopulation displaying the highest clonogenic response. This indicates that the presence of a higher LD amount was an intrinsic feature of the cells and may represent a selective advantage which might allow resistant cells to survive damages, induced by ROS production following exposure to ionizing radiation.

Many intracellular mechanisms participate in ROS production and the Fenton reaction is one of them. In this reaction, ferrous ion is used as a catalyst to convert $H_2O_2$ into the highly oxidative hydroxyl radical (OH•). Iron is an important player in normal cells because it is involved in many processes, and therefore, its homeostasis is tightly regulated. However, in cancer cells, iron homeostasis is dysregulated, and Ferritin, the protein involved in iron storage, has been shown to be elevated in some tumor tissues, thus suggesting that increased iron storage in cancer cells might contribute to cell survival (*Bystrom et al., 2014*).

Previous studies showed a crucial role for FTH1 in cancer aggressiveness. FTH1-silenced MCF7 and H460, cells acquired a mesenchymal phenotype associated with an epithelial to mesenchymal transition and the activation of the CXCR4/CXCL12 signaling pathway (*Aversa et al., 2017*). However, studies on Ferritin and iron roles in radiotherapy are limited. Naz and colleagues demonstrated an

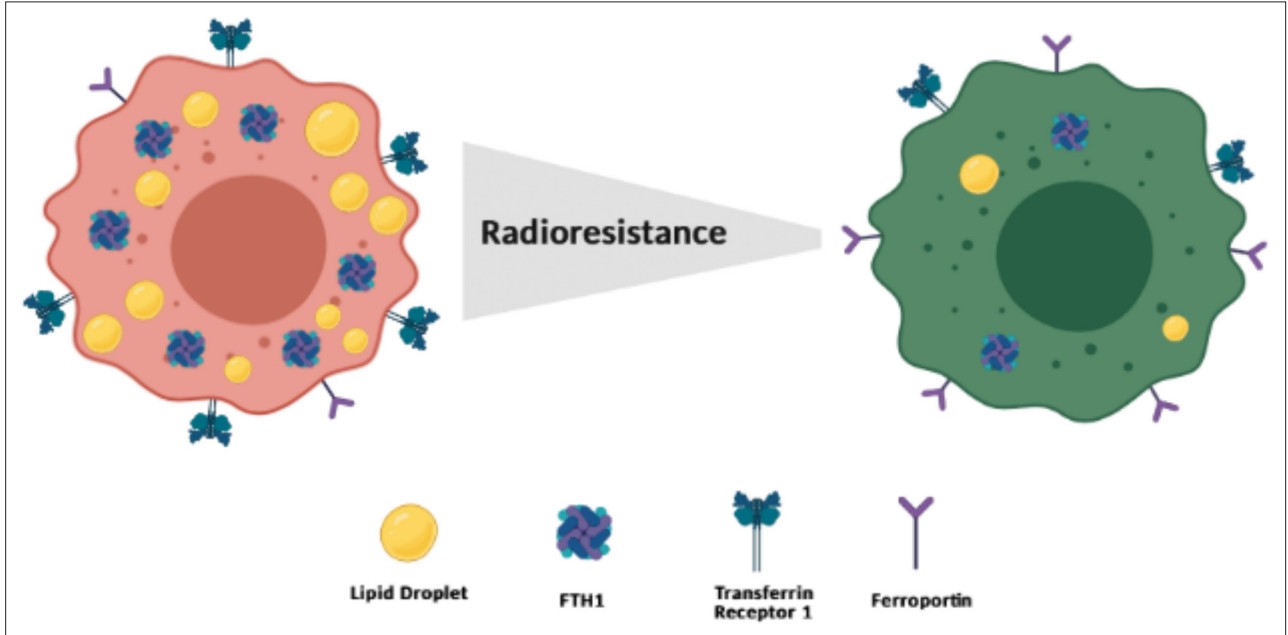

**Figure 5.** Schematic representation of a putative model at the basis of the cancer radioresistance (Created withBioRender.com).

The online version of this article includes the following figure supplement(s) for figure 5:

**Figure supplement 1** Keplan-Meier curves showing the Survival Probability related to the PLIN1-5, FTH1, TFR1 and FPN gene expression.

upregulation of hepatic ferritin and elevated FTL serum levels in sham-irradiated rats (*Wang et al., 2010*).

In this work, we investigated the effects of X-ray radiation on RR cancer cells in order to determine a possible relationship between FTH1 and LDs. We found that surviving cells in all lines showed an upregulation, although at different extents, of FTH1 protein. Moreover, in both MCF7- and H460-LD$^{High}$ fractions, FTH1 resulted upregulated as compared to the MCF7- and H460-LD$^{Low}$ counterparts. The link between FTH1 and LDs was further confirmed by the reduction of LD accumulation in FTH1-silenced cells. Moreover, FTH1 downregulation associated with the downregulation of transferrin receptor that mediates extracellular iron uptake and the upregulation of ferroportin, responsible for iron release, indicating that most likely iron levels in FTH1-silenced cells were unbalanced. In such conditions, cells were significantly more sensitive to ionizing radiation than the relative controls. These findings show a strong correlation between FTH1 expression and LD content in radioresistant cells and, indirectly, suggest that unbalanced intracellular availability of iron produced effects on lipid pathways, mainly on LD accumulation. These data were then corroborated by the overexpression of FTH1 protein in silenced H460 and MCF7 cell lines, where we observed a restored LD content together with an increased clonogenic response. Our findings support the idea that the two cell states (LD$^{High}$/FTH1$^{High}$ and LD$^{Low}$/FTH1$^{Low}$) (*Figure 5*) are not irreversible processes, but they are reversible mechanisms where the big player is the cytoplasmic iron pool.

Alteration in FTH1 expression induces changes of intracellular free Fe levels. Excess iron is cytotoxic, mainly because of the production of ROS, and in our study, cells with reduced ability to store iron also showed reduced RR. However, in the absence of adequate levels of FTH1, treatment with an iron chelator was able to reduce the iron excess inside cells, and this caused a significant LD re-accumulation in both FTH1-silenced (MCF7 and H460) cell lines. Once again, re-established LD content and iron storage resulted in increased RR of both cell lines. Therefore, iron homeostasis is strongly correlated with the surviving ability of the RR cells and LDs are important mediators in these processes.

Although the data reported here need to be validated in more physiologically complex systems, they provide novel insights about LD involvement in the RR of cancer cells and show that this feature is common to different tumor cells analyzed in the present study.

Notably, the FTH1-LD interconnection was also seen in mice ablated for both iron regulatory proteins 1 and 2 (IRP1 and IRP2) using Cre/Lox technology (*Galy et al., 2008*; *Galy, 2010*). In these works, IRP ablation in the hepatic and gastrointestinal tissues strongly enhanced FTH1 expression, induced transferrin downregulation and ferroportin upregulation. Interestingly, in both cases, the iron metabolism impairment caused a severe LD accumulation.

TCGA database investigations about the survival probability (SP) related to the PLIN1-5, FTH1, TFR1, and FPN gene expression in samples derived from patients suffering with bladder, breast, low-grade glioma, lung, and prostate cancers (*Figure 5—figure supplement 1*) provided some evidences in accordance with our results. For example, the Kaplan–Meier survival curves for patients with breast cancer showed that high expression of PLIN5 mRNA (*Figure 5—figure supplement 1*) was significantly correlated with a better SP. Accordingly, our data showed that radioresistant breast cancer cells were characterized by a downregulation of PLIN5. Similar trend was observed for PLIN3 in low-grade glioma patients. On the other hand, for other genes we could not find similar correlations. However, it should be noted that these public data refer to a large population of clinical samples from patients not undergoing to any treatment. Currently, there are not available data to compare our results deriving from radioresistant cell lines with X-ray-irradiated clinical samples.

In conclusion, the functional cross-talks between LDs and iron homeostasis in the context of tumor RR (*Figure 5*) need to be more deeply explored in order to determine the potential contribution of other related pathways and organelles in these processes. This would offer the opportunity for a better understanding of the mechanisms behind radiation responses and may suggest novel strategies for incrementing the radiotherapy curative capacity.

Lastly, a common effort has to be put forth in the identification of robust and functional predictive biomarkers to be used to target the most resistant cancer populations by precise treatments, which need to be as specific as possible for the most tumorigenic cells (CSCs/CICs) while preserving, as much as possible, toxicity on the healthy cell population (*Krause et al., 2017*).

# Materials and methods

**Key resources table**

| Reagent type (species) or resource | Designation | Source or reference | Identifiers | Additional information |
|---|---|---|---|---|
| Cell line (*Homo sapiens*) | MCF-7 | ATCC | Cat# HTB-22 RRID:CVCL_0031 | |
| Cell line (*Homo sapiens*) | H4 | ATCC | Cat# HTB-148 RRID:CVCL_1239 | |
| Cell line (*Homo sapiens*) | H460 | ATCC | Cat# HTB-177 RRID:CVCL_0459 | |
| Cell line (*Homo sapiens*) | T24 | ATCC | Cat# HTB-4 RRID:CVCL_0554 | |
| Cell line (*Homo sapiens*) | PC3 | ATCC | Cat# CRL-1435 RRID:CVCL_0035 | |
| Antibody | (Goat polyclonal) anti-human FTH1 | Santa Cruz Biotechnology | Cat# sc-14416 RRID:AB_2107172 | WB(1:200) |
| Antibody | (Goat polyclonal) anti-human HSC70 | Santa Cruz Biotechnology | Cat# sc-1059 RRID:AB_2120291 | WB (1:2000) |
| Antibody | (Goat polyclonal) anti-mouse IgG-HRP | Santa Cruz Biotechnology | Cat# sc-20550 RRID:AB_631738 | WB (1:2000) |
| Recombinant DNA reagent | pLKO.1 | Addgene (gift from F.S. Costanzo) *Di Sanzo et al., 2011* | | |
| Recombinant DNA reagent | pcDNA3 | Addgene (gift from F.S. Costanzo) *Aversa et al., 2017* | | |
| Chemical compound, drug | CM-H$_2$DCFDA | Thermo Fisher Scientific | Cat# C6827 | |
| Chemical compound, drug | Nile Red | Thermo Fisher Scientific | Cat# N1142 | |
| Chemical compound, drug | Deferoxaminemesylate | Sigma-Aldrich | Cat# D9533 | |
| Chemical compound, drug | Puromycin dihydrochloride | Acros organics | Cat# BP2965 | |
| Chemical compound, drug | Power SYBR Green PCR Master mix | Thermo Fisher Scientific | Cat# 4367659 | |
| Chemical compound, drug | LD540 | Synthetised by Enamine | | |
| Commercial assay or kit | Pierce BCA Protein Assay Kit | Thermo Fisher Scientific | Cat# 23,225 | |
| Commercial assay or kit | High Pure isolation RNA kit | Roche | Cat# 11828665001 | |
| Commercial assay or kit | RT$^2$ First strand Kit (50) | Qiagen | Cat# 330,404 | |
| Commercial assay or kit | Lipofectamine 3000 Transfection Reagent | Thermo Fisher Scientific | Cat# L3000075 | |
| Sequence-based reagent | Custom-made qPCR primers – GAPDH | Sigma-Aldrich | | Forward – 5'-GCATCCTGGGCTACACTGAG-3' Reverse – 5'-AAAGTGGTCGTTGAGGGCA-3' |
| Sequence-based reagent | Custom-made qPCR primers – FTH1 | Sigma-Aldrich | | Forward – 5'-CATCAACCGCCAGATCAAC-3 Reverse –5'-GATGGCTTTCACCTGCTCAT-3' |

*Continued on next page*

Continued

| Reagent type (species) or resource | Designation | Source or reference | Identifiers | Additional information |
|---|---|---|---|---|
| Sequence-based reagent | Custom-made qPCR primers – TfR1 | Sigma-Aldrich | | Forward – 5'-CTGGTAAACTGGTCCATGCT-3' Reverse – 5'-GTGATTTTCCCTGCTCTGAC-3' |
| Sequence-based reagent | Custom-made qPCR primers – FPN | Sigma-Aldrich | | Forward – 5'-GGTGTCTGTGTTTCTGGT-3' Reverse – 5'-GTCTAGCATTCTTGTCCAC-3' |
| Sequence-based reagent | Custom-made qPCR primers – CD24 | Sigma-Aldrich | | Forward – 5'-CCTGTCAGAGCTGTGTGGAC-3' Reverse – 5'-GCTGGGTAGAGTGGTGTGT-3' |
| Sequence-based reagent | Custom-made qPCR primers – CD44 | Sigma-Aldrich | | Forward – 5'-GGGTTCATAGAAGGGCACGT-3' Reverse – 5'-GGGAGGTGTTGGATGTGAGG-3' |
| Sequence-based reagent | Custom-made qPCR primers – CD133 | Sigma-Aldrich | | Forward – 5'-AAGCATTGGCATCTTCTATGG-3' Reverse – 5'-AGAGAGTTCGCAAGTCCTTG-3' |
| Sequence-based reagent | Custom-made qPCR primers – CD166 | Sigma-Aldrich | | Forward – 5'-CGATGAGGCAGACGAGATAAG-3' Reverse – 5'-TAGACGACACCAGCAACAAG-3' |
| Sequence-based reagent | Custom-made qPCR primers – ALDH1 | Sigma-Aldrich | | Forward – 5'-AACTGGAATGTGGAGGAGGC-3' Reverse – 5'-ATGATTTGCTGCACTGGTCC-3' |
| Sequence-based reagent | Custom-made qPCR primers – PLIN1 | Sigma-Aldrich | | Forward – 5'-GACAAGGAAGAGTCAGCCCC-3' Reverse – 5'-GAGAGGGTGTTGGTCAGAGC-3' |
| Sequence-based reagent | Custom-made qPCR primers – PLIN2 | Sigma-Aldrich | | Forward – 5'-ACAGGGGTGATGGACAAGAC-3' Reverse – 5'-ATCATCCGACTCCCCAAGAC-3' |
| Sequence-based reagent | Custom-made qPCR primers – PLIN3 | Sigma-Aldrich | | Forward – 5'-CACCATGTTCCGGGACATTG-3' Reverse – 5'-GCACCTGGTCCTTCACATTG-3' |
| Sequence-based reagent | Custom-made qPCR primers – PLIN4 | Sigma-Aldrich | | Forward – 5'- GTTCCAGGACCACAGACA-3' Reverse – 5'CCTACACTGAGCACATCC-3' |
| Sequence-based reagent | Custom-made qPCR primers – PLIN5 | Sigma-Aldrich | | Forward – 5'-GATCACTTCCTGCCCATGAC-3' Reverse – 5'-GCTGTCTCCTCTGATCCTCC-3' |
| Sequence-based reagent | Custom-made qPCR primers – SOD1 | Sigma-Aldrich | | Forward – 5'-GCAGATGACTTGGGCAAAGG-3' Reverse – 5'-TGGGCGATCCCAATTACACC-3' |
| Sequence-based reagent | Custom-made qPCR primers – SOD2 | Sigma-Aldrich | | Forward – 5'-CTGGAACCTCACATCAACGC-3' Reverse – 5'-CCTGGTACTTCTCCTCGGTG-3' |
| Sequence-based reagent | Custom-made qPCR primers – GPX1 | Sigma-Aldrich | | Forward – 5'-CCCAAGCTCATCACCTGGTC-3' Reverse – 5'-TGTCAATGGTCTGGAAGCGG-3' |
| Sequence-based reagent | Custom-made qPCR primers – Catalase | Sigma-Aldrich | | Forward – 5'-CGTGCTGAATGAGGAACAG-3' Reverse – 5'-GACCGCTTTCTTCTGGATG-3' |

## Cell cultures and transfection

MCF7 human breast adenocarcinoma and H4 neuroglioma cell lines (ATCC) were cultured in DMEM medium (Thermo Fischer Scientific) supplemented with fetal bovine serum (FBS) 10 % (Thermo Fischer Scientific), Pen/Strep 1 % (Thermo Fischer Scientific). H460 human non-small lung cancer cells (ATCC) were cultured in RPMI 1640 (Thermo Fischer Scientific) medium supplemented with 10 % FBS and 1 % penicillin–streptomycin (Thermo Fischer Scientific). T24 bladder carcinoma cell line (ATCC) was cultured in McCoy's medium (Thermo Fischer Scientific) supplemented with FBS 10 % (Thermo Fischer Scientific), Pen/Strep 1 % (Thermo Fischer Scientific), and Hepes 1 % (Thermo Fischer Scientific). PC3 prostate adenocarcinoma cells (ATCC) were cultured in F-12K medium (Thermo Fischer Scientific), supplemented with FBS 10 % (Thermo Fischer Scientific) and Pen/Strep 1 % (Thermo Fischer Scientific). All these cell lines were maintained at 37 °C in a humidified 5 % $CO_2$ atmosphere and cultured following ATCC recommendations.

Lentiviral transduced MCF7 and H460 were kindly provided by the laboratory headed by Prof. Francesco Saverio Costanzo at the University Magna Graecia of Catanzaro. Both cell lines were stably transduced with a lentiviral DNA containing either an shRNA that targets the 196–210 region of the FTH1 mRNA (sh29432) (MCF-7shFTH1, H460shFTH1) or a control shRNA without significant homology to known human mRNAs (MCF-7shRNA, H460shRNA). MCF-7shRNA and MCF-7shFTH1

were cultured in DMEM medium (Thermo Fischer Scientific) supplemented with FBS 10 % (Thermo Fischer Scientific), Pen/Strep 1 % (Thermo Fischer Scientific), and puromycin 1 µg/mL (Sigma-Aldrich). H460shRNA and H460shFTH1 were cultured in RPMI 1640 (Thermo Fischer Scientific) medium supplemented with 10 % FBS and 1 % penicillin–streptomycin (Thermo Fischer Scientific), puromycin 1 µg/mL. All cell lines were maintained at 37 °C in a humidified 5 % $CO_2$ atmosphere.

All cell lines used in the manuscript have been authenticated by means of Multiplex human Cell line Authentication (MCA) test and analyzed for mycoplasma contamination by EZ-PCR Mycoplasma Test Kit (Biological Industries).

### Radiation treatment and clonogenic assay

Irradiation has been carried out using a Multi Rad 225kV irradiator. Cells, seeded at a density of 3.5 × $10^5$ and 1.0 × $10^6$ cells for 0 and 6 Gy, respectively, were irradiated with 6 Gy at room temperature and left in culture for 72 hrs in order to get only surviving cells at the end of the culturing time. Fresh medium was replaced every day.

Cell survival was evaluated using a standard colony forming assay. H4 LD$^{High}$ and LD$^{Low}$, H460 LD$^{High}$ and LD$^{Low}$, MCF7 LD$^{High}$ and LD$^{Low}$, PC3 LD$^{High}$ and LD$^{Low}$, T24 LD$^{High}$ and LD$^{Low}$, H460 shRNA, H460 shFTH1, H460 shFTH1+ DFO, H460 shFTH1/pcDNA$_3$, H460 shFTH1/pcDNA$_3$FTH1, MCF7 shRNA, MCF7 shFTH1, MCF7 shFTH1+ DFO, MCF7 shFTH1/pcDNA$_3$, and MCF7 shFTH1/pcDNA$_3$FTH1 populations were collected soon after sorting. Cells were seeded into six-well plates (Corning) at a density of 2 × $10^2$–1 × $10^4$ cells/well, irradiated (2, 4, and 6 Gy single dose) with a Multi Rad 225kV irradiator and incubated for 7–12 days at 37 °C in a humidified atmosphere with 5 % $CO_2$. Following incubation, colonies were fixed in 100 % ethanol and stained using a 0.05 % crystal violet solution. Only the colonies with more than 35 cells were counted. Surviving fractions were calculated after correction for plating efficiency of control cells. At least three independent experiments, each in duplicate, have been performed for the above-mentioned cell samples.

### Cell sorting

T24, MCF7, H460, H4, PC3 cell suspensions were washed in phosphate-buffered saline (PBS) (Thermo Fischer Scientific). Cells were then stained with LD540 for 10 min at 37 °C in the incubator. The excess of dye was washed away with PBS, and cells were resuspended in sorting buffer (PBS Ca/Mg-free, BSA 0.5%, EDTA 2 mM, and Hepes 15 mM).

Cells were sorted in two populations (LD$^{High}$ and LD$^{Low}$) using a FACSAria Fusion (BD Bioscience). Sorting gates were established based on the 10 % most bright and 10 % most dim subpopulation.

All cell sorting experiments have been carried out within 1 hr upon sorting to avoid that sorted cells could start becoming heterogeneous again.

### LD staining

Depending on the project needs, LD content was assessed by staining cells with two different dyes: LD540 and Nile Red. For FACS measurements, 4 × $10^5$ cells have been harvested, washed with PBS, and then stained with 0.1 µg/mL LD540 or 1/500 (from a saturated stock solution in acetone) Nile Red. Stained cells were analyzed at the FACS Canto II (BD Bioscience). Instead, for the confocal microscopy analysis, 4 × $10^3$ cells have been cultured on a 35 mm Glass Bottom Dishes (MatTek Life Science) and then fixed with PFA 4 %. After washing out the PFA, fixed cells were stained with 0.1 µg/mL LD540 and 1 µg/mL Hoechst 33,342 (Thermo Fischer Scientific). Cells were imaged by a Leica SP5 or a Zeiss LSM710 confocal microscope systems.

### ROS staining

Intracellular ROS content was measured by freshly prepared chloromethyl dichlorodihydrofluorescein diacetate (CM-H$_2$DCFDA, Thermo Fisher Scientific) dye resuspended in anhydrous dimethyl sulfoxide (Thermo Fisher Scientific). Briefly, 4 × $10^5$ cells were collected and washed three times with PBS Ca$^+$/Mg$^+$-free 1× and soon after incubated with 3.5 µM CM-H$_2$DCFDA in pre-warmed Hank's balanced salt solution (HBSS, Thermo Fischer Scientific) for 20 min at 37 °C, in the dark. The samples were analyzed, after having washed them with PBS, by using a FACSCanto II flow cytometer (BD Biosciences).

### LD and ROS co-staining

4 × $10^5$ cells were harvested, washed with PBS 1×, and soon after stained with 1/500 (from a saturated stock solution in acetone) of Nile Red for 20 min at 37 °C in the dark. Stained cells were washed three

times and then incubated with 3.5 µM of CM-H$_2$DFCDA in HBSS for 20 min at 37 °C in the dark. After one wash in PBS 1× , cells were analyzed using a FACS Canto II (BD Bioscience).

## Antibodies and western blot analysis

H4 0 and 6 Gy, H460 0 and 6 Gy, MCF7 0 and 6 Gy, PC3 0 and 6 Gy, T24 0 and 6 Gy, MCF7 shRNA, MCF7 shFTH1/pcDNA$_3$, MCF7 shFTH1/pcDNA$_3$FTH1, H460 shRNA, H460 shFTH1/pcDNA$_3$, and H460 shFTH1/pcDNA$_3$FTH1 cells were washed twice with cold PBS and incubated for 20 min with 300 µL of 1× Ripa Buffer (Cell Signaling) additioned with HaltTM Protease Inhibitor Single-Use Cocktail (Thermo Fisher Scientific) and HaltTM Phosphatase Inhibitor Single-Use Cocktail (Thermo Fisher Scientific), both diluted 1:100. Cells were then transferred to tubes and, after centrifugation at 14,000 × g at 4 °C for 20 min, the supernatants were collected. Protein concentration was measured by BCA Protein assay kit (Thermo Fisher Scientific) at 562 nm using BSA to produce a standard curve. For protein analysis, 15 µg of whole-cell extracts for each sample were electrophoresed under reducing condition in 10 % SDS–polyacrylamide gels and then electrophoretically transferred onto PVDF membrane filters (Bio-Rad Laboratories), using Trans-Blot Turbo Transfer System (Bio-Rad Laboratories, Hercules, CA). In order to prevent the non-specific antibody binding, blots were blocked for 1 hr with BSA blocking buffer, 5 % in PBS, with 0.1 % Tween-20 (TWEEN 20 Bio-Rad Laboratories). Membranes were washed with PBS-0.1% Tween and incubated with antibodies in blocking solution overnight at 4 °C. Antibody used was a rabbit anti H- ferritin (1:200; Santa Cruz Biotechnology, Dallas, TX). PBS-0.1% Tween-20 was used to remove the excess of primary antibody and then the membranes were incubated in blocking solution with goat anti-mouse IgG-HRP (1:2000, Santa Cruz Biotechnology) secondary antibody. Subsequently, blots were rinsed with 0.1 % PBS-Tween and developed with Clarity Western ECL Substrate (Bio-Rad Laboratories) using Amersham Imager 680. Protein levels were analyzed by ImageJ 1.52 p software.

## RNA isolation and real-time PCR (qRT-PCR)

Total RNA was isolated from 6 Gy irradiated and non-irradiated cells, LD$^{High}$ and LD$^{Low}$ sorted cells, MCF7 shRNA and MCF-7 shFTH1 as well as H460 shRNA and H460 shFTH1 using the High Pure RNA isolation kit (Roche) according to the manufacturer´s instructions. All the RNA samples were treated with DNase-1 to remove any contaminating genomic DNA and the RNA purity was checked spectroscopically. Then, 1 µg of purified RNA was reverse transcribed using RT 2 First Strand Kit (Qiagen) according to the manufacturer´s instructions.

Gene expression analysis was assessed by Real-Time PCR (qRT-PCR) using the cDNA obtained from the cell samples above reported.

Twenty nanograms of cDNA was amplified in 15 µL of reaction mix containing Power SYBR Green PCR Master mix (ThermoFisher Scientific), 20 pmol of each primer pair and nuclease-free water on a StepOne Plus System (ThermoFisher scientific). The thermal profile consisted of 1 cycle at 95 °C for 10 min followed by 40 cycles at 95 °C for 15 s, 60 °C for 1 min. Relative gene expression was normalized to that of the gene encoding the human GAPDH which served as an internal control. Data analysis was performed using the 2-ΔΔCt method.

## Widefield and confocal microscopy

T24, H4, PC3, MCF7, MCF7 shRNA, MCF7 shFTH1, H460, H460 shRNA, and H460 shFTH1 were seeded and stained with LD540 as reported in the Lipid Droplet Staining section. Zeiss LSM710 and Leica SP5 microscopes, both equipped with a 40× and 63 ×, were used to image LDs.

## Image analysis

Z-stack images of LD540-stained cells were taken using a Leica SP5 confocal-laser-scanning microscope equipped with a 40× oil immersion i-Plan Apochromat (numerical aperture 1.40) objective. LD540 were visualized using the 488 nm line of an Argon laser and a 505–530 nm BP filter. Twelve-bit images were acquired and post processed for the LD quantification. Briefly, the background was subtracted using ImageJ's Rolling ball radius tool. The images were further processed with Gaussian filter, thresholded, and segmented with Find Maxima tool. Finally, images were analyzed with Analyze Particles tools. All the image processing was performed automatically with constant settings using in-house developed macro for Fiji generously provided by Dr. Damir Krunic.

Student's t-test with unequal variances was used for the calculation of statistical significances. Differences of two groups with p-values below 0.05 were considered statistically significant.

## FTH1 reconstitution

MCF7 shFTH1 and H460 shFTH1 cells were seeded in six-well plates at $3 \times 10^5$ cells/well and grown overnight prior to transfection.

All plasmids were transfected with Lipofectamine 3000 transfection reagent (Thermo Fisher Scientific) following manufacturer's instructions. FTH1 reconstitution was performed using 2.5 µg/µl of the expression vector containing the coding sequence of human FTH1 cDNA (pcDNA3/FTH1) (MCF-7 shFTH1/pcDNA$_3$FTH1 and H460 shFTH1/pcDNA$_3$FTH1), while 2.5 µg/µl of pcDNA$_3$ plasmid was used as negative control (MCF-7 shFTH1/pcDNA$_3$ and H460 shFTH1/pcDNA$_3$). Transfection efficiency was tested by western blot and qPCR after 48 hrs. All transfection experiments were repeated in triplicate.

## Deferoxamine treatment

MCF-7-Wt, MCF-7-shRNA, MCF-7-shFTH1, H460-Wt, H460-shRNA, and H460-shFTH1 cells were seeded in 100 mm$^2$ petri dishes (Corning) at a concentration of $1.5 \times 10^6$ cells/plate containing 10 mL of DMEM or RPMI-1640 (supplemented with 10 % FBS) and incubated for 24 hrs. Then, cells were treated with 50 µM DFO (deferoxaminemesylate salt). Cells cultured in normal medium were used as control. After 24 hrs of treatment, cells were collected and used for ROS and LD detection.

Analysis of the association between SP and PLIN1-5, FTH1, TFR1, and FPN genes in breast, bladder, low-grade glioma, prostate, and lung cancer patients.

The association between the SP of individual PLIN1-5, FTH1, TFR1, and FPN genes in breast, bladder, low-grade glioma, prostate, and lung cancer samples from patients was explored by Kaplan–Meier plotter on the TCGAportal. Briefly, each of the genes above mentioned was, respectively, entered into the database in order to get the Kaplan–Meier SP plots. All different cancer patients were divided into the 'low' and 'high' expression groups based on the mRNA expression levels of the individual gene. Afterward, the Kaplan–Meier method, together with the log-rank test, was used for univariate SP analysis. A p-value of <0.05 was considered to be statistically significant.

## Statistics

All data here presented are shown as mean values ± SD of the irradiated or 'treated' samples relative to the untreated control. Statistical and data analysis was carried out using GraphPad Prism nine software. Statistical differences between treated and untreated samples were assessed by T-test and one-way ANOVA. The threshold for statistical significance was set to p=0.05.

# Acknowledgements

We gratefully acknowledge the Imaging and FACS facilities of the DKFZ for their precious support. LT has received funding from AIRC and from the European Union's Horizon 2020 Research and Innovation Programme under the Marie Skłodowska-Curie grant agreement no 800,924.

# Additional information

## Funding

| Funder | Grant reference number | Author |
| --- | --- | --- |
| AIRC | 800924 | Luca Tirinato |

The funders had no role in study design, data collection and interpretation, or the decision to submit the work for publication.

## Author contributions

Luca Tirinato, Conceptualization, Data curation, Formal analysis, Funding acquisition, Methodology, Project administration, Resources, Supervision, Writing - original draft, Writing - review and editing; Maria Grazia Marafioti, Joana Filipa Guerreiro, Data curation, Formal analysis, Methodology; Francesca

Pagliari, Data curation, Formal analysis, Investigation, Methodology, Writing - original draft, Writing - review and editing; Jeannette Jansen, Geraldine Genard, Data curation, Methodology; Ilenia Aversa, Rachel Hanley, Investigation, Methodology; Clelia Nisticò, Methodology; Daniel Garcia-Calderón, Formal analysis; Francesco Saverio Costanzo, Project administration, Resources, Supervision; Joao Seco, Conceptualization, Data curation, Funding acquisition, Methodology, Project administration, Supervision, Writing - review and editing

### Author ORCIDs

Luca Tirinato  http://orcid.org/0000-0001-9826-2129
Francesca Pagliari  http://orcid.org/0000-0002-5547-222X
Jeannette Jansen  http://orcid.org/0000-0002-8625-3978
Rachel Hanley  http://orcid.org/0000-0002-2627-1146
Clelia Nisticò  http://orcid.org/0000-0002-0787-9527
Geraldine Genard  http://orcid.org/0000-0002-9495-0335
Joana Filipa Guerreiro  http://orcid.org/0000-0003-1960-603X
Joao Seco  http://orcid.org/0000-0002-9458-2202

### Decision letter and Author response

Decision letter https://doi.org/10.7554/eLife.72943.sa1
Author response https://doi.org/10.7554/eLife.72943.sa2

## Additional files

### Supplementary files
• Transparent reporting form
• Source data 1. Source data for *Figure 3* and *Figure 4*.

### Data availability
All data generated or analyzed during this study are included in the manuscript and supporting files. All Source data files have been provided.

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
