## [Decision Letter]

[Editors' note: this paper was reviewed by Review Commons.]

**Acceptance summary:**

This is a very interesting paper that uncovers lipid droplet (LD) content of tumor cells derived from different types of cancer as impacting radio-sensitivity. They suggest that tumor cell heterogeneity in LD is pre-existing and selected for and that iron imbalance and FTH1 are key mediators. They show that silencing of FTH1 enhances radiosensitivity. This paper has broad interest in the field of cancer.

---

## [Author Response]

Reviewer #1 (Evidence, reproducibility and clarity (Required)):In this work Tirinato and co-Authors used different experimental approaches to trace correlations between cancer cell stemness, lipid droplets, iron homeostasis and radiation resistance. Their findings were acquired by using different cancer cell lines from different origin, and using diverse techniques, including cytometry, microscopy, clonogenic assays and shRNAi.In general, the manuscript is well written and organized. It is also easy to be followed and the authors managed to convince the readers about the importance of this important aspect of cancer metabolism. The fact that cell lipid droplets content might condition cancer survival to radiotherapy poses novelty and therefore it deserves attention in the delimitation of new anticancer therapies or protocols.There are however some issues that should be amended to improve the quality of the manuscript.Major comments:1) The main concern is that all the work is based on cancer cell lines. The use of some cell lines derived from clinical samples or analysis of the clinical data already deposited in bank webs could be useful to support their conclusions. This last could be easy. Exploration of this depositories could help the author to reinforce the correlation between the expression of the lipid droplets genes, as well as that related with the iron metabolism, with the radiotherapy efficacy in patients.

We thank and agree with the Reviewer for this important point. In order to reinforce the correlation between LDs, FTH1 and cancer radioresistance we have explored several gene databases and studied, more in dept, the literature.

Nevertheless, since the topic is quite new and many of our gene analysis results have been carried out on irradiated cells, surviving the treatment, with the aim at understanding gene modifications induced by the radiation itself, it resulted pretty hard comparing our results with the information present in the online databases. No data are currently available from clinical samples collected from X-ray treated patients. However, we followed the Reviewer's suggestion and included a new Figure (Figure 5—figure supplement 1) in the Supplementary Information which reports the Survival Probability for the genes analyzed in the paper (PLIN1-5, FTH1, TFR1 and FPN) in the respective five cancer types: bladder, breast, lung, glioma and prostate cancers. These data have been derived from the TCGAportal.

We have added a brief sentence at the end of the Discussion and a description of the Figure S5 as reported below:

“TCGA database investigations about the Survival Probability (SP) related to the PLIN1-5, FTH1, TFR1 and FPN gene expression in samples derived from patients suffering with bladder, breast, low grade glioma, lung and prostate cancers (Figure S5) provided some evidences in accordance with our results. For example, the Keplen-Meier survival curves for patients with breast cancer showed that high expression of PLIN5 mRNA (Figure S5) was significantly correlated with a better SP. Accordingly, our data showed that radioresistant breast cancer cells were characterized by a downregulation of PLIN5. Similar trend was observed for PLIN3 in low grade glioma patients. On the other hand, for other genes we could not find similar correlations. However, it should be noted that these public data refer to a large population of clinical samples from patients not undergoing to any treatment. Currently, there are not available data to compare our results deriving from radioresistant cell lines with X-ray irradiated clinical samples.”

Moreover, a new section describing the Method used for the database investigation was added to Material and Methods, as reported below:

“Analysis of the Association Between Survival Probability and PLIN1-5, FTH1, TFR1 And FPN Genes in Breast, Bladder, Low Grade Glioma, Prostate and Lung Cancer Patients

The association between the Survival Probability (SP) of individual PLIN1-5, FTH1, TFR1 and FPN genes in breast, bladder, low grade glioma, prostate and lung cancer samples from patients was explored by Kaplan-Meier Plotter on the TCGAportal. Briefly, each of the genes above mentioned was respectively entered into the database in order to get the Kaplan-Meier survival probability plots. All different cancer patients were divided into the ‘low’ and ‘high’ expression groups based on the mRNA expression levels of the individual gene. Afterward, the Kaplan-Meier method together with the Log-rank test were used for univariate SP analysis. A p-value of <0.05 was considered to be statistically significant.”

Furthermore, thanks to the Reviewer' comment, we have found two very important papers ("Iron Regulatory Proteins Are Essential for Intestinal Function and Control Key Iron Absorption Molecules in the Duodenum – Cell Metabolism 2007" and "Iron Regulatory Proteins Secure Mitochondrial Iron Sufficiency and Function – Cell Metabolism 2010" both published by Galy et al.) where the FTH1-LD interconnection in mice, ablated for both iron regulatory proteins 1 and 2 (IRP1 and IRP2) using Cre/Lox technology, is clearly reported. In this regard, we have briefly described these evidences in the Discussion section, as reported below:

"Notably, the FTH1-LD interconnection was also seen in mice ablated for both iron regulatory proteins 1 and 2 (IRP1 and IRP2) using Cre/Lox technology by B. Galy ^23,24^. In these works, IRP ablation in the hepatic and gastrointestinal tissues strongly enhanced FTH1 expression, transferrin downregulation and ferroportin upregulation. Interestingly, in both cases, the iron metabolism impairment caused a severe LD accumulation.”

2) When analyzed in detail I have some comments regarding specific sections:Lipid droplets detection images (Figures1, 3 y 4). Why are the nuclei size look so different in both conditions?

We would like to thank the Reviewer for the question. Radiation treatment affects profoundly cell size and shape, mainly at the nuclear level. In this regard, one of the first changes detectable after irradiation is the nuclear swelling (Arthur C. Upton, The nucleus of the cancer cell: effects of ionizing radiation, Experimental Cell Research, 1963). The prevalence of these changes and their time of onset depend on the radiosensitivity of the investigated cells and on the irradiation conditions. Our approach was to set the radiation dose (6 Gy) and the post-exposure recovery time (72 hrs) up fixed for all the different cell lines. These, in turn, affected the nuclear size in a cell-line dependent manner.

The scale bar for all images was set up at 20 μm.

3) FTH1 expression. Figure 3A vs 3C and 4A. As depicted is difficult to have a clear picture of the variations in expression in FHT1 in the different cell lines.

We agree with the Reviewer and for this reason we decided to plot the FTH1 differences also by bar charts (new Figure 3B). This should help the Readers to better understand the FTH1 WB quantitative differences among the cell samples.

4) Why the authors evaluate the success of shRNAi by PCR if the WB works so well?

We thank the Reviewer for the comment. We measured the success of the FTH1 downregulation by Real-Time PCR to ensure that it was effectively achieved at a mRNA level. Then, we confirmed the functional downregulation at a protein level by western blot. In fact, a protein reduction without a corresponding mRNA reduction may suggest that the silencing is mediating its effects at the translation level (Doench JG, Peterson CP, Sharp PA (2003) siRNAs can function as miRNAs, Genes Dev 17:438-42).

5) Is there any correlation between the RR and the expression of FTH1 intra cell lines?

This is a very good point. We did not run any experiment to figure it out, but we strongly believe that there might be a similar correlation between RR and FTH1, like we observed with LDs. This is due to the fact that the LD numbers and FTH1 are tightly connected each other. So, the hypothesis that there might be a FTH1 gradient within the same cell sample and this, in turn, could be proportional to cancer radioresistance is something very plausible.

Further investigations will take into consideration what the Reviewer is suggesting.

6) What happen when FTH1 is downregulated in the rest of the cell lines? More than restore, the authors overexpressed FTH1, what is the result when FTH1 is overexpressed in the different cell lines?

We performed the FTH1 silencing only in H460 and MCF7 cell lines. We did not silence FTH1 in the other samples. The decision was made based on three main reasons: *i*) both breast and lung tumors are the most commonly diagnosed cancers worldwide (11.7% and 11.4% of the total worldwide cancer cases, respectively) (Sung H. et al., Global Cancer Statistics 2020: GLOBOCAN Estimates of Incidence and Mortality Worldwide for 36 Cancers in 185 Countries, CA Cancer J Clin, 2021, 71: 209-249); *ii*) in both cancers, radiotherapy is one of the main therapeutical approaches currently used in the clinic; *iii*) these two cell lines are widely characterized and largely used for breast and lung cancer modelling.

Minor comments:7) The correlation between nile red staining and ROS is not clear (Figure S2). The authors may try to graphic the ROS mfi of different subpopulations (lineal in X) vs the mfi of red nile (y, in log scale).

We have revised the Figure 1—figure supplement 2 in order to make it more understandable. We have also tried to plot the ROS mfi in a linear scale vs the Nile Red in a log scale

However, the final result does not change, but the graph looks less clear, so we believe that keeping both values in a log scale is better in order to clearly separate the two populations.

8) Abbreviation in introduction ER, is endoplasmic reticulum?

It was corrected accordingly.

9) A picture with a putative model could be helpful to summarize the findings.

We thank the Reviewer for her/his suggestion. We have included in the manuscript a picture (Figure 5) to summarize our findings (see below).

Reviewer #1 (Significance (Required)):As stated above the idea that lipid droplets and iron metabolism might be determinants in the cancer survival to radiotherapy poses novelty and therefore it deserves attention in the delimitation of new anticancer therapies or protocols.Although I have experience in cancer lipid metabolism, I am not an expert in the field of lipid droplets.Reviewer #2 (Evidence, reproducibility and clarity (Required)):Summary:In the study, the authors found that cancer cells including breast, bladder, lung, neuroglioma, and prostate display an increase of Lipid Droplet (LD) after 6 Gy x-ray (Figure 1). And, the cells containing high LDs showed more radioresistance than the cells with low LDs after irradiated with a dose Gy x-ray (Figure 2). Ferritin Heavy Chain (FTH1), the main intracellular iron storage protein, is found to be upregulated after 6 Gy exposure or in the LDs high cells. FTH1 knockdown decreases LD accumulation and increases radiation sensitivity (Figure 3). Overexpression of FTH1 or DFO (an iron chelator agent) treatment in shFTH1 cells rescue the LD accumulation and cancer radioresistance (Figure 4)Major comments:1). The conclusion has some conflict with some publication (PMC5928893) which shows fatty acid oxidation not LDs lead to cancer radioresistance. So the authors should rule out this possibility through knockdown of CPT1.

We thank the Reviewer for her/his comment. The paper reported in the Reviewer's comment refers to nasopharyngeal carcinoma and, even if it is relevant in the field, it describes a different cancer system. Unfortunately, at the best of our knowledge, there are not so many papers showing and/or reinforcing what reported in the PMC5928893. Moreover, the experimental design was different from ours. In the paper, the Authors delivered a dose of 4 Gy (with a dose rate of 2 Gy/min) and analyzed the results after 24 h. In our experiments, we delivered 6 Gy X-rays and analyzed cellular responses after 3 days. This was because we were interested in investigating the connections between FTH1 and LDs in the surviving cells, which represent the most radioresistant population. Therefore, different results can derive from these differences. We cannot exclude that at earlier time points the results might be different, but, as reported above, our goal was to look at the radioresistant cells.

Secondly, LDs, as storage of fatty acids in the form of neutral triacylglycerols, are involved in the fatty acid oxidation. It is postulated that the usual pathway for LD degradation involves their dynamic association with mitochondria where a flow of fatty acids from LDs is directed to the mitochondrial matrix for β-oxidation. Possibly, their use as well as their accumulation can, at some point and under precise stimuli, simultaneously proceed.

Therefore, we feel that the knockdown of CPT1, if on one hand, might provide information on the influence of this enzyme on LD turnover/accumulation, on the other hand it is, at the moment, beyond the scope of our work.

2) Lipid droplets are dynamic in cells. The sorted cells (10% highest or lowest LDexpressing cells) in Figure 3 may not stand for subpopulation, so the authors should add exogenous lipids or cholesterol to test cancer cell radioresistance.

We agree with the Reviewer about the LD dynamism in cells. We have seen that after 24 hrs from the cell sorting the LD^High^ and LD^Low^ subpopulations turn again heterogeneous. We are sorry to disagree with the Reviewer about external stimuli. Adding exogeneous lipids or cholesterol to increase the number of LDs within cancer cells would have meant to strongly perturb the whole cell metabolism. This was not our aim. For this reason, we have decided to sort the heterogeneous population in the 10% lowest and highest LD-expressing cells. Immediately after the sorting, we have irradiated the two populations and run the clonogenic assays. This is the only way, in our opinion, to preserve the homogeneity of the two LD subpopulations (at the time of the treatment) without affecting their metabolism exogenously.

3) It is impossible to overexpress FTH1 in shFTH1 cells (the stable shRNA will target all mRNA of FTH1) (Figure 4 and methods section: cell culture and FTH1 Reconstitution).

We agreed with the Reviewer when she/he says that “It is impossible to overexpress FTH1 in shFTH1 cells”, because in principle the silencing of FTH1 by stable shRNA will degrade the complementary mRNA for FTH1 and thus no protein expression would be expected. However, in our experiment, we used a plasmid to overexpress FTH1, which, based on the results, was able to overcome this effect, as we could see the overexpression of FTH1 in shFTH1/pcDNA3FTH1 cells. We do not know the reason of this evidence, but we can hypothesize a couple of possibilities. We used a vector containing the FTH1 cDNA designed on the coding sequence (CDS) (Ref. https://www.ncbi.nlm.nih.gov/nuccore/NM_002032.2) to re-express the protein. The initial silenced cell lines were transduced with a lentiviral DNA containing a shRNA that targets the 196–210 region of the FTH1 mRNA (sh29432). Instead, the CDS for the FTH1 mRNA, used to re-express the protein, is included in the region “236…787”. Thus, it might be that this sequence is not targeted by the shRNA. In other words, the reintroduced complete CDS might be not targeted for the degradation, thus allowing the cell’s machinery to translate the CDS of the mRNA. Alternatively, it might be that the amount used of pcDNA3FTH1 to transfect the silenced cells was high enough to increase the gene copy number for the FTH1 at such a level that the degradation process was overcome making the protein overexpression possible and detectable.

4) The relationship between the free cytoplasmic iron and LD accumulation is not so convincing. Add exogenous iron to test LD accumulation.

We thank the Reviewer for this consideration. Actually, we report that the availability of cytoplasmic iron is correlated with LD content, not by directly measuring the amount of iron, but assuming that silencing FTH1 will cause a modulation of citoplasmic iron. Besides the fact that adding iron to growing cells is a complex matter and many parameters can influences the iron solution chemistry, the overload of free iron in the cells is toxic and in fact iron increases are often reported to induce apoptosis.

More importantly, our main interest was to modulate the amount of FTH1.

Of course, the Reviewer's suggestion is a very interesting point in a more general approach where cytoplasmic iron pool, and by consequence its related proteins, is the main player of the study. In our case, instead, we wanted to show that the number of LDs was connected to the FTH1 expression.

However, in preliminary experiments we tried to measure the free cytoplasmic iron and discriminate between Fe2+ and Fe3+ by magnetometry employing a superconducting quantum interference device (SQUID) without success.

Minor comments:5) Remove Figure 1C which is not related to the conclusion.

We thank the Reviewer, but we have decided to keep it into the main text because we have added in the Supplementary Information a new Figure (Figure 5—figure supplement 1) showing the Survival Probability for the genes analyzed in the paper for all five tumors (as suggested by the other Reviewer). Moreover, since PLINs are involved in the LD formation and structure, their analysis could strengthen the evidence about the ionizing radiation effect onto the LD metabolism.

6) Figure 2 label error: LD520 should be LD540.

We have corrected the mistake.

7) Figure 3, A: change loading control HSC70 which not so stable in the cells; D: add quantification of LD number.

We thank the Reviewer for her/his observations.

As far as the stability of HSC70 is concerned, we chose this loading control because it was more stable than GAPDH and β−Actin. In fact, these last two common controls were strongly affected by radiation and were cell type-dependent, while HSC70 levels fluctuated less, as it is possible to see from the Figures reported in the manuscript.

Moreover, LD quantification for the analyzed cell lines was added to the Figure 3 and 4.

8) Figure 4, A: change loading control HSC70, and repeat western of MCF7 shFTH1/pcDNA3.

We thank the Reviewer for her/his observations.

We have repeated the WB of MCF7 shFTH1/pcDNA3 and added the WB quantification (Figure 4A, panel A and B).

As already reported above (Answer 3), we did not change the loading control HSC70 because it was the most stable among all the controls we have tested so far.

9) Line 111, "…that general ROS levels resulted not altered…" should be "…that general ROS levels were not altered…"

We corrected it accordingly to the Reviewer suggestion.

10) Figure S4 legend: "Figure S2" should be "Figure S4".

We corrected it accordingly to the Reviewer suggestion.

Reviewer #2 (Significance (Required)):The FTH1 affects Lipid Droplets is novel (some results in this study have published: radiation led to LD accumulation (PMC5928893) and an increase of FTH1 (PMC4688087 and PMID:32937103)).The finding is helpful to improve radiation therapy which may combine with drugs targeting FTH1 or iron metabolism.The researchers who worked in cancer treatment are interested in this finding.My expertise is cancer lipid metabolism and cancer therapy.